# 3D Numerical Analysis of the Natural Ventilation Behavior in a Colombian Greenhouse Established in Warm Climate Conditions

**Edwin Villagran** [1,*]**, Rommel Leon** [2]**, Andrea Rodriguez** [1] **and Jorge Jaramillo** [3]

[1]  Corporación Colombiana de Investigación Agropecuaria—Agrosavia, Centro de Investigación Tibaitatá—Km 14, vía Mosquera-Bogotá, Mosquera 250040, Colombia; arodriguezr@agrosavia.co

[2]  Corporación Colombiana de Investigación Agropecuaria—Agrosavia, Centro de Investigación Caribia—Sevilla, Magdalena 478020, Colombia; rleon@agrosavia.co

[3]  Corporación Colombiana de Investigación Agropecuaria—Agrosavia, Centro de Investigación La Selva—Rionegro, Antioquia 054040, Colombia; jejaramillo@agrosavia.co

\*  Correspondence: evillagran@agrosavia.co; Tel.: +57-1-4227-300 (ext. 1239)

**Abstract:** Global food production and availability in hot climate zones are limited by biotic and abiotic factors that affect agricultural production. One of the alternatives for intensifying agriculture and improving food security in these regions is the use of naturally ventilated greenhouses, an alternative that still requires information that allows technical criteria to be established for decision-making. Therefore, the objective of this work was to study the spatial distribution of temperature and relative humidity inside a greenhouse built in the Colombian Caribbean. The methodological approach included the implementation of an experimentally validated 3D numerical simulation model. The main results obtained allowed to determine that the airflows generated inside the greenhouse had average velocities below 0.5 m/s and were mainly driven by the thermal effect of natural ventilation. It was also found that the gradients generated between the interior of the structure and the exterior environment presented values lower than 2.0 °C for temperature and −6.3% for relative humidity. These values can be considered low in comparison with other structures evaluated in other regions of the world where the gradients can reach values higher than 10 °C and 13% for temperature and relative humidity, respectively.

**Keywords:** numerical simulation; airflow; temperature; relative humidity

## 1. Introduction

By the year 2050, the world's population is expected to increase by some 3700 million people to a total of 9700 million, of which approximately 68% are expected to live in cities [1–3]. This aspect is combined with accelerated anthropogenically induced climate change, the associated environmental crisis, the continuing loss of natural resources and food insecurity in some populations worldwide, mainly in developing countries such as those in the tropical region of Latin America and the Caribbean [4–7]. They create a number of current and future challenges for society, including increasing the resilience and reducing the vulnerability of agricultural food production systems [8,9], establish strategies for sustainable intensification of agriculture in order to provide food of high nutritional quality for the population [10–12], and at the same time reduce poverty rates by increasing farmers' incomes [13].

One of the alternatives for the intensification of agricultural production systems in open fields is the production in greenhouses of the so-called passive type. This type of structure is widely used in the main regions of plant production worldwide, such as Asia and a number of countries on the

Mediterranean coast [14,15], its use allows the production of high yield and high quality horticultural crops during all seasons [16]. In these types of greenhouses, the management of the microclimate is carried out through natural ventilation, which is an economic and sustainable alternative that allows to control the thermal excesses and humidity inside these structures, and additionally it is the only way of carbon enrichment of the air inside the greenhouse [17,18]. The driving forces of natural ventilation are generated from airflow movements created by a physical phenomenon such as natural convection via buoyancy, known as the thermal effect, and by pressure differences created between the indoor and outdoor environment from the opening of the side and roof ventilation areas, known as the wind effect [19–22].

The generation of an adequate microclimate for the growth and development of crops is highly related to the efficiency of climate control that can be performed by natural ventilation, which in turn is dependent on characteristics such as design, size, and geometric configuration of the greenhouse, width and number of attached buildings, shape of the roof, arrangement and size of the ventilation areas, the magnitude of the temperature differential between the interior and exterior environment of the greenhouse, the presence and type of crop, the protection or not of the ventilation areas with insect-proof porous screens, and finally, by the direction, speed, and intensity of the wind from the exterior environment [23–29].

Worldwide, there are a considerable number of greenhouse designs that have been designed for the specific climatic, social, cultural, and economic conditions of each territory [30,31]. Therefore, the selection of the appropriate design for each region is a quite complex task that requires a serious analysis by decision-makers, farmers, and even the builders themselves, since the selection or design of an inappropriate greenhouse model can generate that the plants are exposed to inadequate temperature and humidity conditions, subjecting them to a condition of stress that can cause a reduction in production, the appearance of diseases, and even the death of the plant, factors that undoubtedly generate an economic impact on producers and an unsustainable production system [17,32,33].

Designers of passive greenhouses for hot and humid regions seek to maximize the efficiency of natural ventilation in order to avoid the use of mechanical ventilation for cooling and dehumidification, which would generate additional costs for acquiring equipment and environmental costs associated with the energy requirement of operating such equipment [23,34,35]. The phenomenon of natural ventilation can be quantified and evaluated under different methodologies; a great part of them were exposed and explained by Akrami et al. [10]. For design purposes, it is useful to have an agile and precise tool that allows the evaluation of the airflows generated inside a greenhouse structure and its cooling efficiency under the specific local climatic conditions of each study site [17,21].

One of the most used tools in the last two decades and that, according to the current computational development, allows the integration of all aspects related to the natural ventilation of greenhouses and its relationship with the generation of microclimate through heat transfer and mass interactions, is computational fluid dynamics (CFD), which has been successfully used and allows the study of airflows and their spatial distribution inside greenhouses through 2D and 3D simulation models in steady-state or spatial-temporal distribution through transient or pseudo-transient simulations [29,36–44].

The CFD methodology allows numerical evaluations of microclimate behavior and airflows in different types of greenhouse structures through a simulation approach. This type of approach allows studies to be carried out in a short period of time and under scenarios that are not built on a real scale. It is also possible to vary the operating conditions or ventilation configuration of the greenhouse and the input variables of the numerical model by adjusting them to the climate conditions of the region of study [10,21]. Microclimate variables, such as temperature and relative humidity in passive greenhouses like the one evaluated in this research, depend directly on the efficiency of airflows and the dynamic behavior of the climatic conditions of the outside environment during daylight hours, a compartment that is characteristic of intertropical climate regions.

In the tropical regions of Latin America and the Caribbean and in some countries at other latitudes, the use of greenhouses for agricultural production is limited to high mountain regions where cold and mild climatic conditions predominate [45]. In contrast, in hot climate regions the use of passive greenhouses has failed due to the establishment of structures with designs not adapted to the local climate conditions. This has meant that, in these low-altitude regions, protected agriculture has not developed significantly, allowing it to be a technological tool to mitigate the adverse effects generated by extreme weather conditions and the pressure of insect pests on crops established in the open field.

Therefore, the main objective of this research was to evaluate the thermal and hygrometric behavior and airflows of a new passive greenhouse prototype, designed and built for horticultural production under the climatic conditions of a hot climate region. The evaluation approach considered the use of a validated CFD-3D simulation model, and the results obtained in this research could be used or adapted by producers or decision-makers in other tropical or subtropical climate regions where the design of this type of greenhouses still presents great engineering challenges.

## 2. Materials and Methods

### 2.1. Description of the Greenhouse

For the validation and subsequent application of the CFD model, experimental measurements were carried out in a naturally ventilated flat roof greenhouse (PG). The floor area under the roof was 500 m$^2$, corresponding to longitudinal and transversal dimensions of 20 m and 25 m, respectively (Figure 1). The PG greenhouse was built in the Caribbean region of Colombia, in the municipality of Seville, department of Magdalena, at the Caribbean Research Center in The Corporación Colombiana de Investigación Agropecuaria—AGROSAVIA (longitude: 74°10′ W, latitude: 10°47′ W and altitude: 18 mamsl). With sides and a maximum height of 9.8 m above the ridge in the central part, the PG was equipped with side and front windows with an opening of 2.5 m, with which the side ventilation area is 225 m$^2$, which is equivalent to 45% of the covered floor area. On the other hand, the ventilation areas were complemented by four ventilation regions in the PG roof area, two with a maximum opening of 1.54 m and another two with a maximum opening of 0.98 m, with which the roof ventilation area is 100 m$^2$, which is equivalent to 20% of the covered floor area. All ventilation areas were covered with an insect-proof screen with a thread density (threads cm$^{-1}$ × threads cm$^{-1}$) of 16.1 × 10.2.

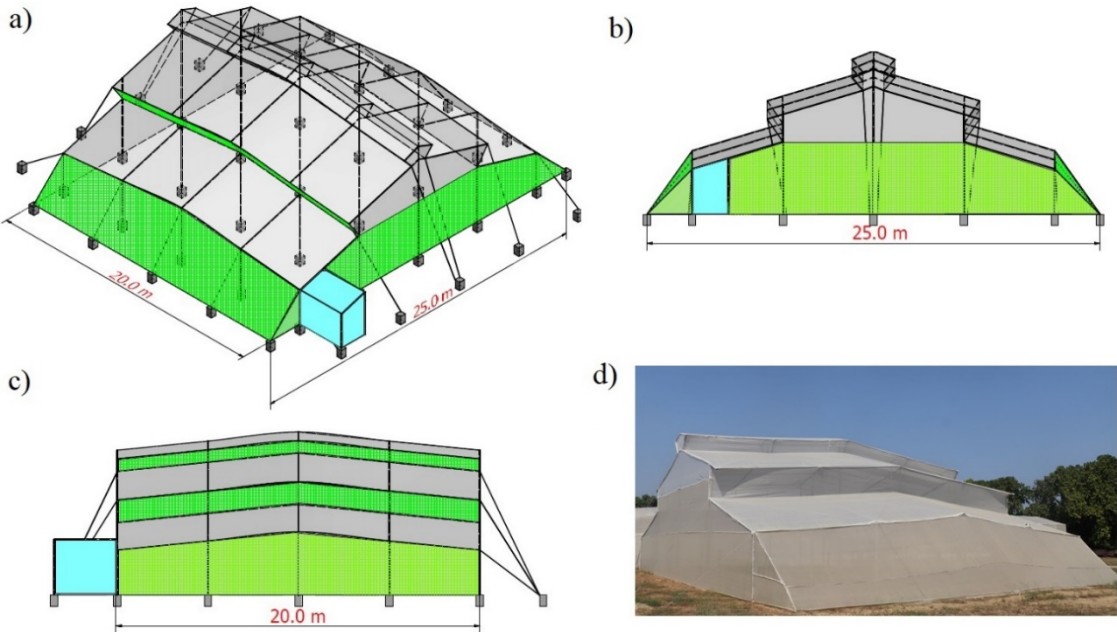

**Figure 1.** Overall dimensions of the naturally ventilated flat roof greenhouse PG, (**a**) Isometric view, (**b**) Front view, (**c**) Side view, and (**d**) Real prototype.

## 2.2. Numerical Model and Governing Equations

Numerical CFD models are a technique for solving a set of non-linear partial differential equations known as Navier Stokes (RANS) averaged by Reynolds, which are discretized through numerical solving methods using a system of linear equations. The CFD methodology allows to calculate the airflow patterns and the temperature and humidity distribution patterns generated inside a greenhouse; the transport phenomena via free convection can be described by Equation (1), which is considered the general transport equation for a fluid in stationary state.

$$\frac{\partial(u_\varnothing)}{\partial x} + \frac{\partial(v_\varnothing)}{\partial y} + \frac{\partial(w_\varnothing)}{\partial z} = \Gamma \nabla^2 \varnothing + S_\varnothing \tag{1}$$

where $y$, $x$, and $z$ represent the coordinates in Cartesian space, $u$, $v$, and $w$ are the components of the velocity vector, $\nabla^2$ is the Laplacian operator, $\Gamma$ is the diffusion coefficient, $\varnothing$ represents the concentration of the quantity transported in a dimensional form (moment, mass, and energy), and $S_\varnothing$ is the source term. The turbulence was incorporated into the numerical model through the standard empirical $k$-$\varepsilon$ model, a model based on the transport equations that solve kinetic turbulent energy and the dissipation of this energy per unit volume $\varepsilon$. This model has been the most used and widely validated in greenhouse airflow studies showing to be efficient with the use of computational resources and providing realistic solutions [46,47]. Equations (2) and (3) are the transport equations for $k$ and $\varepsilon$.

$$\frac{\partial}{\partial x}(\rho k) = \frac{\partial}{\partial x_j}\left[\left(\mu + \frac{\partial k}{x_j}\right)\frac{\partial k}{\partial x_j}\right] + G_k + G_b - \rho\epsilon - Y_M \tag{2}$$

$$\frac{\partial}{\partial t}(\rho\varepsilon) = \frac{\partial}{\partial x_i}\left[\left(\mu + \frac{\mu_t}{\sigma}\right)\frac{\partial\epsilon}{\partial x_i}\right] + \rho C_1 S_\epsilon - \rho C_2 \frac{\epsilon^2}{k + \sqrt{v\epsilon}} + C_{1\epsilon}\frac{\epsilon}{k}C_{3\epsilon}G_b k \tag{3}$$

where $\mu$ is viscosity and $\mu_t$ is turbulent viscosity in (kg/m s), $G_b$ is turbulent kinetic energy generation due to buoyancy, $G_k$ is the generation of turbulent kinetic energy due to velocity gradients, $\sigma_k$ y $\sigma_\epsilon$ are Prandtl's turbulent numbers for $k$ and $\varepsilon$, $v$ is the coefficient of kinematic viscosity, $Y_M$ is the fluctuating expansion in turbulence due to the overall dissipation rate, and $C_{1\epsilon}$, $C_{2\epsilon}$, $C_\mu$, $\sigma_k$ y $\sigma_\epsilon$ are constant with empirically determined values and set by default in the simulation software [48]. Likewise, the effect of air buoyancy caused by the force of gravity and air density changes were added to the momentum equation as a source term by Boussinesq's approximation [49,50]. The presence of insect-proof screens covering the side, front, and roof ventilation surfaces of the structure can be modelled as porous mediums [51]. Taking into account its non-linear moment loss coefficient ($Y$) and its permeability as a function of screen porosity ($K$), the airflow through these screens is governed by Equation (4), describing Darcy-Forcheimer's law.

$$\frac{\partial p}{\partial x} = \frac{\mu}{K}u + \rho\frac{Y}{\sqrt{K}}u|u| \tag{4}$$

where $u$ is the air velocity (m), $\mu$ is the dynamic fluid viscosity (kg/m s), $\rho$ is the density of air (kg/m³), and $\partial x$ the thickness of the porous material (m); the factors $K$ and $Y$ are determined through experimental equations that can be reviewed in the study of Valera et al. [52]. The aerodynamic coefficients requested by the CFD model were established for the type of insect screen used, which is similar to that reported in the study developed by Flores-Velazquez et al. [53]. The species transport equation was activated in order to study the humidity distribution inside PG. This model allows to solve the conservation equation in a turbulent flow for the diffusion, convection, and reactions of the mass fractions for each species defined by the user in the numerical model [54,55]. The mass fraction of the elements can be predicted by the CFD model from Equation (5).

$$\nabla(\rho Y_i) = -\nabla J_1 + R_i + E_i \tag{5}$$

where E is the moisture source term, J is the species diffusion flow, Ri is the species production rate per component i, and Y is the local mass fraction of each species through the solution of the convection–diffusion equation. The radiation model selected was the discrete ordinate (DO) model with angular discretization. This model has been widely used in these types of studies applied to greenhouses and allows for calculating the radiation and convective exchanges that occur in the computational domain, treating the greenhouse roof as a semi-transparent medium [56,57]. Equation (6) describes the DO radiation model.

$$\nabla\left(I_\lambda\left(\underset{r}{\Rightarrow}, \underset{s}{\Rightarrow}\right)\underset{s}{\Rightarrow}\right) + (a_\lambda + \sigma_s)I_\lambda\left(\underset{r}{\Rightarrow}, \underset{s}{\Rightarrow}\right) = a_\lambda n^2 \frac{\sigma T^4}{\pi} + \frac{\sigma_s}{4\pi}\int_0^{4\pi} I_\lambda\left(\underset{r}{\Rightarrow}, \underset{s}{\Rightarrow}'\right)\Phi\left(\underset{s}{\Rightarrow} \cdot \underset{s}{\Rightarrow}'\right)d\Omega' \quad (6)$$

where $I_\lambda$ is the intensity of radiation at a wavelength; $\underset{r}{\Rightarrow}$, $\underset{s}{\Rightarrow}$ are the vectors that indicate the position and direction, respectively; $\underset{s}{\Rightarrow}'$ is the direction vector of the scatter; $\sigma_s$, $a_\lambda$ are the coefficients of dispersion and spectral absorption; $n$ is the index of refraction; $\nabla$ is the divergence operator; $\sigma$ is the Stefan–Boltzmann's constant ($5.669 \times 10^{-8}$ W/m$^2$ K$^4$); $\Phi$, $T$ and $\Omega$ are the phase function, the local temperature (°C), and the solid angle, respectively. Likewise, in order to simplify the resolution of the 3D CFD model, no type of crop was included, this in order to speed up the numerical calculation and establish the behavior of the airflow and temperature under the worst possible scenario, which means conditions where there is no presence of plants, a scenario where a large part of the radiation incident on the interior of the greenhouse is converted into heat, which generates an increase in the temperature of the interior air. To solve the pressure-momentum coupled equations, a semi-implicit solution method was adopted for pressure-linked equations through a second-order discretization scheme for momentum, energy, and turbulence, the convergence criteria for these variables were set at $10^{-6}$ [57,58].

### 2.3. Computational Domain and Meshing Process

The phases that comprise the definition of the computational domain size, the definition of boundary conditions, and the size and type of grid are key stages in ensuring the acceptable prediction of airflow over the target building or construction [59]. The computational domain had a size of 313 m, 63 m, and 308 m for the X, Y, and Z axes, respectively. This size was defined in relation to the maximum height of the greenhouse H = 9 m; therefore, the perimeter borders were defined at a distance of 16H and the upper border at a distance of 7H, as recommended by several numerical studies of natural ventilation of buildings [27,59–61]. To determine the size of the numerical grid in order to guarantee solution independence, adequate convergence, and acceptable computational performance, a total of 8 grids of different cell sizes were evaluated within the range {950,876–22,345,679}. The selected grid was number 4, which contains a total of 10,675,918 square elements formed in an unstructured grid (Figure 2).

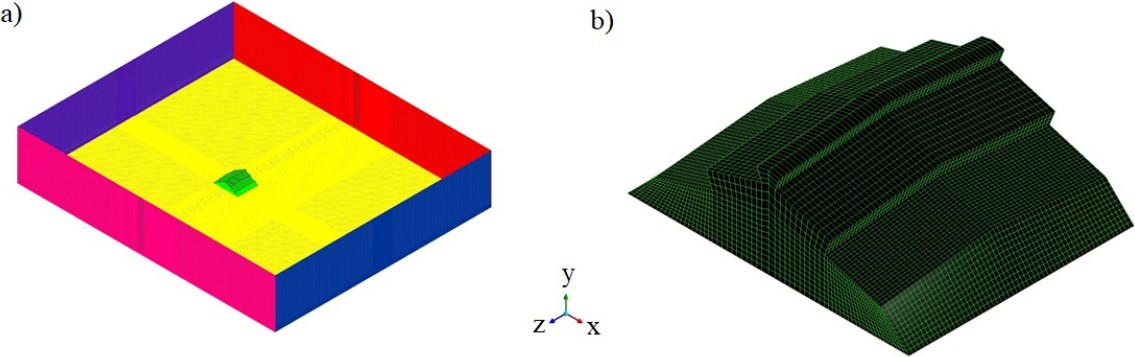

**Figure 2.** (**a**) Computer domain and (**b**) Meshing of the PG greenhouse.

The evaluation process of the numerical grid independence test was carried out according to the one developed by Villagran et al. [29]. The mean value of the airflow velocity, temperature, and relative humidity inside the PG greenhouse were selected as the main parameters according to which the independence analysis was performed. This analysis includes the qualitative evaluation of the value of these parameters as a function of the mesh size under the same simulation conditions. The selection of the grid size is made once it is observed that the evaluation parameters tend to have a constant behavior independently of the numerical grid size (Figure 3). The quality of the grid was evaluated using the orthogonality criterion, for which 98.2% of the cells were in the high-quality range [62,63].

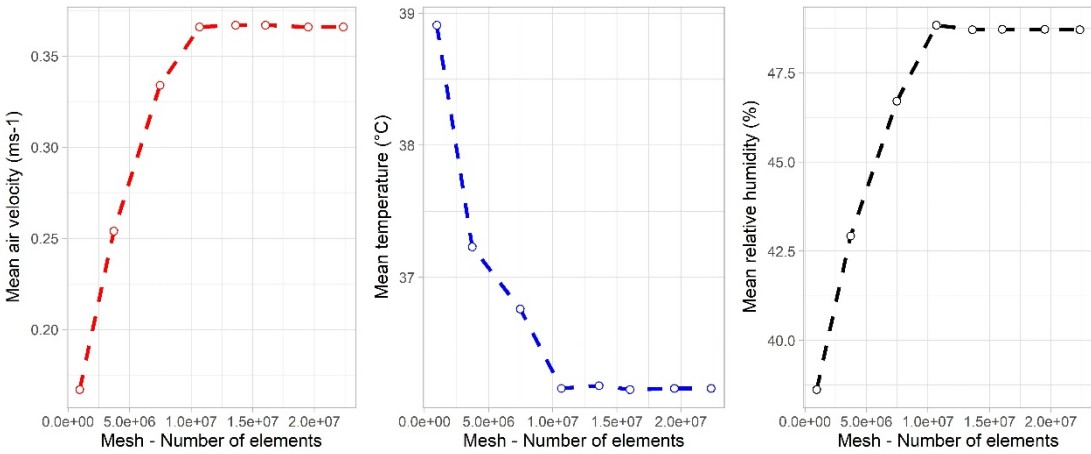

**Figure 3.** Behavior of the variables analyzed in the numerical grid independence test.

The boundary conditions established to the computational domain were of symmetrical properties for the sides parallel to the airflow direction; to the airflow exit limit a pressure outlet condition was established, while for the air entry limit a logarithmic profile was established according to Richard and Hoxey [64], establishing the soil roughness parameters according to the conditions of the study site and applying them to the numerical model as explained in King et al. [65]. For the greenhouse roof and floor areas of the computer domain, wall conditions were established, while for the areas where the porous insect screen is located, a limit of porous medium was established. In each of these regions, the optical and physical properties of each of the materials were defined and are summarized in Table 1.

**Table 1.** Physical and optical properties used in computational fluid dynamics (CFD) simulation [29,51].

| Physical and Optical Properties of the Materials | | | | |
|---|---|---|---|---|
| | **Air** | **Ground** | **Polyethylene** | **Porous Screen** |
| Density ($\rho$, kg m$^{-3}$) | 1.225 | 1.400 | 920 | 910 |
| Thermal conductivity ($k$, W/m °K) | 0.0242 | 1.5 | 0.33 | 0.31 |
| Specific heat ($Cp$, J/°K kg) | 1006.43 | 1.738 | 2600 | 1.800 |
| Coefficient of thermal expansion (1/K) | 0.0033 | | | |
| Absorptivity | 0.19 | 0.9 | 0.06 | 0.2 |
| Scattering coefficient | 0 | −15 | 0 | 0 |
| Refractive index | 1 | 3 | 1.53 | 0.05 |
| Emissivity | 0.9 | 0.95 | 0.70 | 0.45 |

*2.4. Validation of the CFD Model and Simulated Scenarios*

During the period from 1 January 2020 to 30 March 2020, climate information was recorded and stored in the greenhouse's external and internal environment at a measurement frequency of every 10 min. In the outdoor environment, a weather station was used, Davis-Vantage 2 plus 6162 (Davis Instruments, Hayward, CA, USA). On the other hand, inside the PG greenhouse (Figure 4), at a height

of 1.7 m above ground level (*y*-axis), were 5 micro weather stations of the WatchDog 1000 (Spectrum Technologies, Aurora, IL, USA).

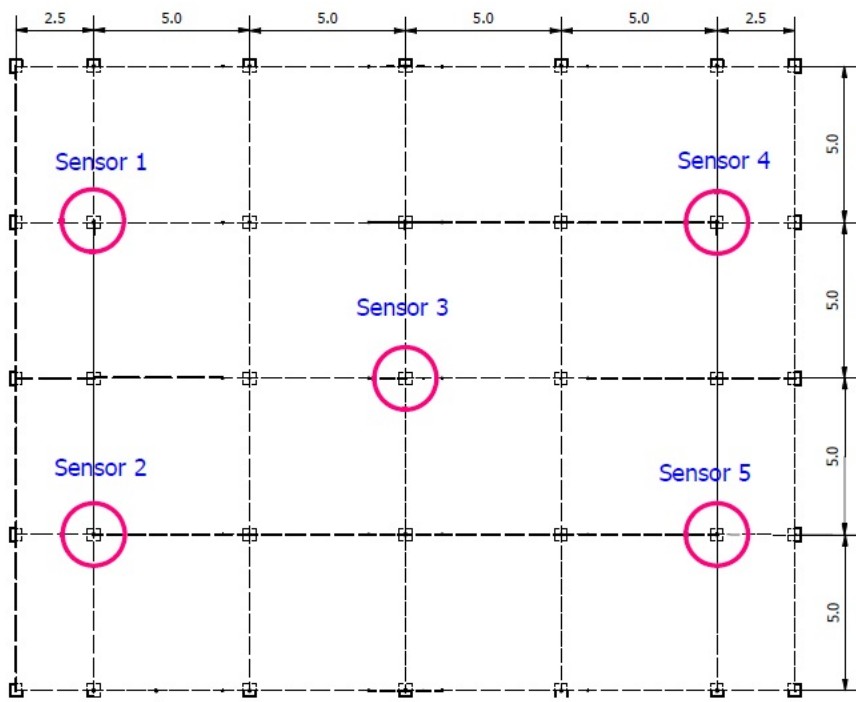

**Figure 4.** Distribution of the measuring sensors inside the PG greenhouse.

The climatic information collected during the measurement period in the outdoor environment such as air temperature and humidity, solar radiation, and wind direction and speed were averaged for each hour, and the values obtained for the period between 7:00 and 17:00 were established as initial conditions for each hour (Table 2).

**Table 2.** Average hourly climatic conditions for the region of study.

| Hour | Temperature (°C) | Relative Humidity (%) | Solar Radiation (W/m²) | Wind Speed (m/s) | Wind Direction |
|------|------------------|-----------------------|------------------------|------------------|----------------|
| S01—Hour 7 | 24.9 | 81.4 | 46.6 | 0.2 | WNW |
| S02—Hour 8 | 27.8 | 75.6 | 177.8 | 0.2 | WNW |
| S03—Hour 9 | 30.1 | 68.6 | 376.7 | 0.2 | W |
| S04—Hour 10 | 31.9 | 62.4 | 580.8 | 0.4 | W |
| S05—Hour 11 | 33.5 | 56.6 | 713.7 | 0.7 | W |
| S06—Hour 12 | 34.8 | 53.1 | 792.1 | 1.01 | W |
| S07—Hour 13 | 35.2 | 52.2 | 802.7 | 1.22 | W |
| S08—Hour 14 | 35.4 | 50.8 | 719.4 | 1.31 | W |
| S09—Hour 15 | 34.9 | 50.2 | 547.6 | 1.13 | N |
| S10—Hour 16 | 34.1 | 51.8 | 347.9 | 0.72 | N |
| S11—Hour 17 | 33.1 | 54.6 | 184.6 | 0.51 | N |

Also, in order to evaluate the validity of the numerical model, once the hourly simulations were completed, temperature and relative humidity data were extracted for each sampling point within PG (Figure 4). Initially it was proceeded to check the validity or the rejection of the numerical simulation model through a statistical analysis by means of a hypothesis test for the difference of measures with homogeneous variances. This analysis was complemented by a comparison of the simulated data with the data measured through goodness-of-fit parameters such as mean absolute error (MAE) Equation (7),

root-mean-square error (RMSE) Equation (8), and finally through the coefficient of determination ($R^2$) Equation (9).

$$MAE = \frac{1}{N} \sum_{i=1}^{N} |Dm_i - Ds_i| \tag{7}$$

$$RMSE = \sqrt{\sum_{i=1}^{N} \frac{|Dm_i - Ds_i|^2}{N}} \tag{8}$$

$$R^2 = 1 - \frac{\sum_{i=1}^{N} |Dm_i - Ds_i|^2}{\sum_{i=1}^{N} |Dm_i - \overline{Dm}|^2} \tag{9}$$

where $N$ is the number of samples, $Dm_i$ are the measured values of temperature and relative humidity, $Ds_i$ are the simulated temperature and relative humidity values at a point $I$, and $\overline{Tm}$ is the average of the measured values.

## 3. Results

### 3.1. Data and Model Validation

In Table 3 can be found the summary of temperature and relative humidity data sets obtained through the experimental test and through numerical simulation for each of the measurement points and for each scenario evaluated, data sets that will be purchased to verify the validity of the model. The temperature values increase in magnitude from S01—Hour 7 to scenario S08—Hour 14 and, on the contrary, the humidity values decrease for these same scenarios. Subsequently, the temperature decreases again, and the humidity level recovers its value in magnitude.

**Table 3.** Measured and simulated data for each scenario evaluated.

| Scenario | Sensor | T-M * | T-S * | RH-M * | RH-S * | Scenario | Sensor | T-M * | T-S * | RH-M * | RH-S |
|----------|--------|-------|-------|--------|--------|----------|--------|-------|-------|--------|------|
| S01—Hour 7 | 1 | 25.2 | 25.1 | 78.8 | 79.1 | S07—Hour 13 | 1 | 35.9 | 35.5 | 50.3 | 51.2 |
| S01—Hour 7 | 2 | 25.3 | 25.2 | 78.9 | 79.8 | S07—Hour 13 | 2 | 36.2 | 35.5 | 49.5 | 51.3 |
| S01—Hour 7 | 3 | 26.1 | 25.6 | 75.8 | 76.7 | S07—Hour 13 | 3 | 38.4 | 37.1 | 43.8 | 47.9 |
| S01—Hour 7 | 4 | 25.7 | 25.3 | 76.9 | 77.9 | S07—Hour 13 | 4 | 38.5 | 37.5 | 43.3 | 45.8 |
| S01—Hour 7 | 5 | 25.7 | 25.4 | 77.3 | 78.5 | S07—Hour 13 | 5 | 38.6 | 37.3 | 42.8 | 45.5 |
| S02—Hour 8 | 1 | 28.3 | 28.2 | 73.9 | 74.2 | S08—Hour 14 | 1 | 36.4 | 35.8 | 47.5 | 49.9 |
| S02—Hour 8 | 2 | 28.3 | 28.4 | 74.2 | 74.1 | S08—Hour 14 | 2 | 36.5 | 35.7 | 47.4 | 49.8 |
| S02—Hour 8 | 3 | 29.3 | 28.9 | 69.2 | 70.2 | S08—Hour 14 | 3 | 38.1 | 37.5 | 41.3 | 44.7 |
| S02—Hour 8 | 4 | 28.7 | 28.4 | 69.5 | 71.8 | S08—Hour 14 | 4 | 38.0 | 37.6 | 41.5 | 45.9 |
| S02—Hour 8 | 5 | 29.5 | 29.2 | 67.3 | 69.4 | S08—Hour 14 | 5 | 37.9 | 37.8 | 42.0 | 45.7 |
| S03—Hour 9 | 1 | 30.9 | 30.8 | 67.0 | 67.5 | S09—Hour 15 | 1 | 35.7 | 35.6 | 48.6 | 49.8 |
| S03—Hour 9 | 2 | 31.0 | 30.7 | 66.1 | 67.4 | S09—Hour 15 | 2 | 35.6 | 35.4 | 46.4 | 47.9 |
| S03—Hour 9 | 3 | 32.4 | 31.8 | 58.9 | 62.0 | S09—Hour 15 | 3 | 36.2 | 35.7 | 45.6 | 46.8 |
| S03—Hour 9 | 4 | 31.1 | 30.6 | 64.1 | 65.7 | S09—Hour 15 | 4 | 35.9 | 35.5 | 46.3 | 48.7 |
| S03—Hour 9 | 5 | 32.4 | 31.4 | 63.8 | 64.6 | S09—Hour 15 | 5 | 36.6 | 36.5 | 44.7 | 45.5 |
| S04—Hour 10 | 1 | 32.5 | 32.3 | 60.1 | 61.3 | S10—Hour 16 | 1 | 35.3 | 34.9 | 47.8 | 49.3 |
| S04—Hour 10 | 2 | 32.6 | 32.3 | 59.5 | 61.2 | S10—Hour 16 | 2 | 35.2 | 34.7 | 49.2 | 51.1 |
| S04—Hour 10 | 3 | 34.5 | 33.7 | 54.1 | 55.4 | S10—Hour 16 | 3 | 35.9 | 35.1 | 46.8 | 49.4 |
| S04—Hour 10 | 4 | 33.9 | 33.6 | 55.6 | 56.8 | S10—Hour 16 | 4 | 35.6 | 34.8 | 48.2 | 50.2 |
| S04—Hour 10 | 5 | 34.7 | 33.8 | 54.3 | 56.9 | S10—Hour 16 | 5 | 36.1 | 35.3 | 45.8 | 48.3 |
| S05—Hour 11 | 1 | 34.5 | 34.1 | 54.5 | 55.7 | S11—Hour 17 | 1 | 33.9 | 33.8 | 50.3 | 52.1 |
| S05—Hour 11 | 2 | 34.3 | 34.1 | 55.1 | 55.6 | S11—Hour 17 | 2 | 33,9 | 33.5 | 51.8 | 53.3 |
| S05—Hour 11 | 3 | 36.1 | 35.5 | 50.3 | 51.8 | S11—Hour 17 | 3 | 34.8 | 34.3 | 49.7 | 51.1 |
| S05—Hour 11 | 4 | 36.2 | 35.4 | 48.9 | 51.4 | S11—Hour 17 | 4 | 34.4 | 33.7 | 50.7 | 52.8 |
| S05—Hour 11 | 5 | 36.3 | 35.5 | 50.2 | 52.0 | S11—Hour 17 | 5 | 35.2 | 34.5 | 49.2 | 51.4 |
| S06—Hour 12 | 1 | 36.1 | 35.1 | 48.9 | 52.1 | | | | | | |
| S06—Hour 12 | 2 | 36.0 | 35.1 | 51.2 | 52.0 | | | | | | |
| S06—Hour 12 | 3 | 37.9 | 36.8 | 46.5 | 48.8 | | | | | | |
| S06—Hour 12 | 4 | 37.5 | 37.2 | 46.2 | 46.7 | | | | | | |
| S06—Hour 12 | 5 | 38.1 | 37.3 | 44.9 | 46.5 | | | | | | |

T-M* Measured temperature; T-S* Simulated temperature; RH-M* Measured relative humidity; RH-S* Simulated relative humidity.

In Table 4 can be found the results obtained from the statistical analysis conducted to determine the validity or rejection of the CFD model. Once the normality of the measured and simulated data sets was verified, the hypothesis test was performed to determine if the variances of the data groups are homogeneous. The value of the contrast statistic F was calculated for a significance level of 0.05, obtaining values for F of 1.095 and 1.101, while for *p*-value of 0.750 and 0.724 for temperature and relative humidity, respectively. The *p*-value was much higher than the significance level, which is why the null hypothesis raised cannot be rejected. Therefore, it can be considered that the variances of the data are equal ($\sigma_{(Dm)}^2 = \sigma_{(Ds)}^2$).

Once the equality of the variances has been determined, we proceed to propose and calculate the hypothesis contrast for the difference of the means. For these cases, the null hypothesis was established as $H_0: \mu_{Dm} = \mu_{Ds}$. The *p*-value obtained was 0.4606 and 0.398 for temperature and relative humidity, values that are greater than the significance level of 0.05, so the null hypothesis is accepted. This test also includes the confidence interval for the difference in means, for the case of temperature is {−0.8690, 1.9054} and for humidity it is {−5.8293, 2.3384}; as both intervals include zero, it can be interpreted that, like the hypothesis test, the difference between the means of the observed and simulated data is not significantly different from zero. Therefore, considering that the mean value of the data sets is statistically similar, the simulation model can be accepted.

**Table 4.** Test statistic results for the model.

| F Test to Compare Two Variances | $H_0: \sigma_{(Dm)}^2 = \sigma_{(Ds)}^2$  o  $H_1: \sigma_{(Dm)}^2 \neq \sigma_{(Ds)}^2$ | |
|---|---|---|
| | **Temperature** | **Relative Humidity** |
| F | 1.095 | 1.101 |
| *p*-value | 0.750 | 0.724 |
| 95% confidence interval | {0.636,1.870} | {0.642,1.888} |
| | The null hypothesis ($H_0$) is accepted. | |
| **Two Sample *t*-test** | $H_0: \mu_{Dm} = \mu_{Ds}$  o  $H_1: \mu_{Dm} \neq \mu_{Ds}$ | |
| | **Temperature** | **Relative Humidity** |
| T | 0.740 | −0.847 |
| *p*-value | 0.460 | 0.398 |
| 95% confidence interval | {−0.869,1.905} | {−5.829,2.338} |
| Mean ($\mu$) | $\mu_{Dm} = 33.92\ °C$ $\mu_{Ds} = 33.40\ °C$ | $\mu_{Dm} = 54.77\%$ $\mu_{Ds} = 56.51\%$ |
| | The null hypothesis ($H_0$) is accepted. | |

The quantitative results obtained for the goodness-of-fit parameters between the measured and simulated data (Table 5) showed that, for temperature, the MAE values behaved within the range of 0.24 and 0.94 °C and for RMSE between 0.26 and 1.02 °C. For relative humidity, the MAE values behaved within the range of 0.86 and 3.25%, while for RMSE they were within 0.91 and 3.36%.

**Table 5.** Comparison of measured and simulated temperature and humidity.

| | **Temperature (°C)** | | | **Relative Humidity (%)** | | |
|---|---|---|---|---|---|---|
| | **MAE*** | **RMSE*** | $R^2$ | **MAE** | **RMSE** | $R^2$ |
| S01—Hour 7 | 0.28 | 0.32 | 0.94 | 0.86 | 0.91 | 0.93 |
| S02—Hour 8 | 0.24 | 0.26 | 0.91 | 1.16 | 1.46 | 0.94 |
| S03—Hour 9 | 0.50 | 0.58 | 0.87 | 1.46 | 1.71 | 0.96 |
| S04—Hour 10 | 0.50 | 0.57 | 0.95 | 1.62 | 1.72 | 0.95 |
| S05—Hour 11 | 0.56 | 0.60 | 0.97 | 1.52 | 1.68 | 0.97 |
| S06—Hour 12 | 0.82 | 0.86 | 0.92 | 1.68 | 1.94 | 0.84 |
| S07—Hour 13 | 0.94 | 1.02 | 0.97 | 2.42 | 2.68 | 0.91 |
| S08—Hour 14 | 0.51 | 0.56 | 0.96 | 3.25 | 3.36 | 0.96 |
| S09—Hour 15 | 0.26 | 0.30 | 0.82 | 1.42 | 1.52 | 0.87 |
| S10—Hour 16 | 0.66 | 0.68 | 0.81 | 2.10 | 2.18 | 0.96 |
| S11—Hour 17 | 0.48 | 0.52 | 0.83 | 1.82 | 1.88 | 0.88 |

MAE*: Mean absolute error; RMSE*: Root-mean-square error.

The 1:1 scatter plots and the linear regression curve for a 95% confidence interval were constructed for temperature in each simulated scenario (Figure 5). In general terms, there is good agreement between the measured and simulated data, which is verified with coefficients of determination ($R^2$) ranging from 0.87 to 0.97 (Table 3).

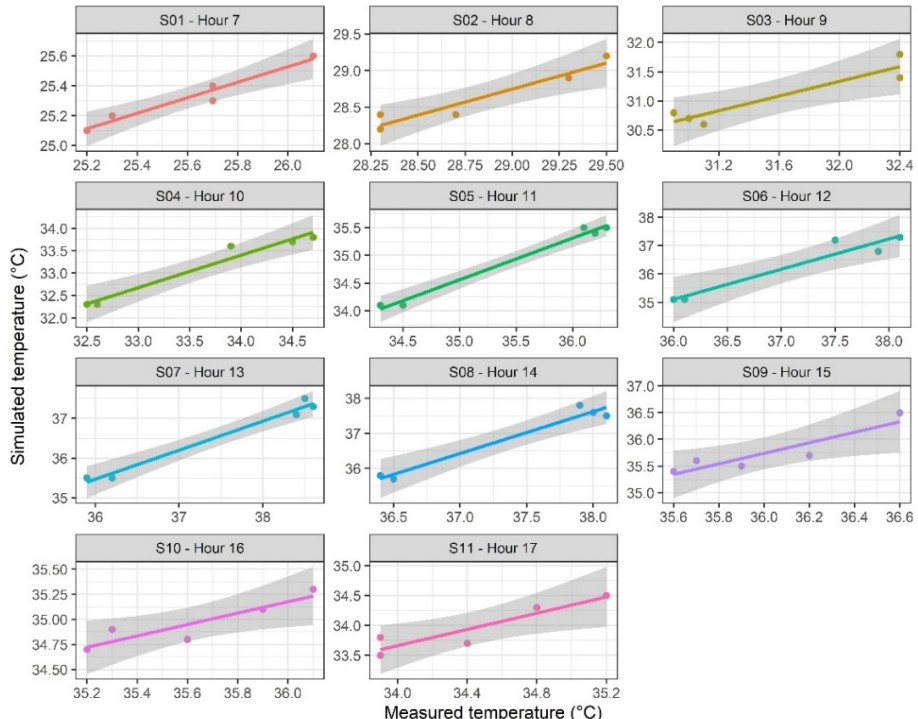

**Figure 5.** Regression curves between measured and simulated indoor temperatures (°C).

This was also done for the relative humidity (Figure 6), for which values of R2 between 0.84 and 0.97 were obtained (Table 3).

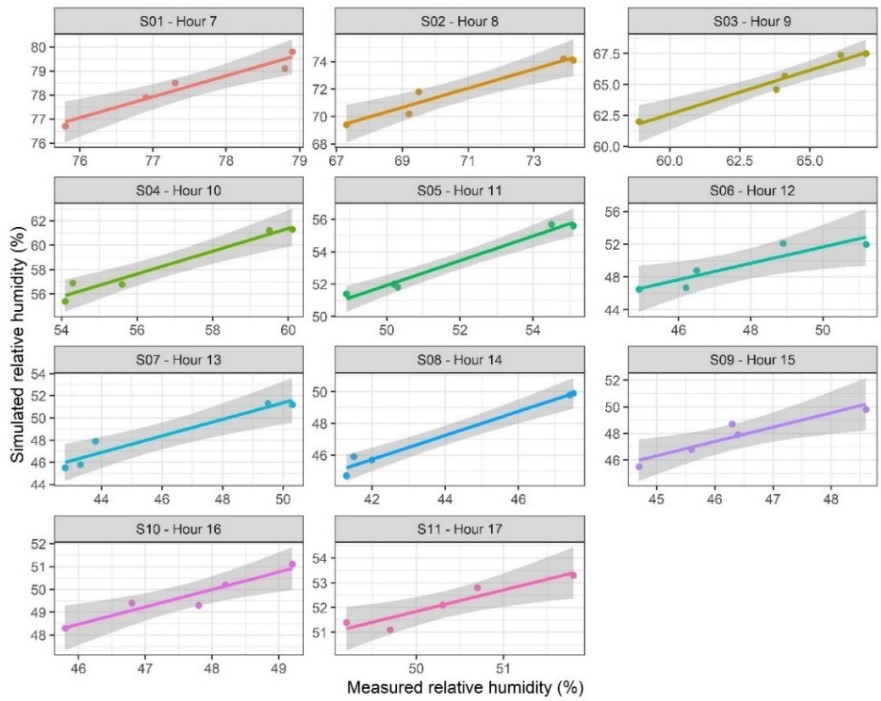

**Figure 6.** Regression curves between measured and simulated indoor relative humidity values (°C).

### 3.2. Airflow Pattern Simulations

The qualitative behavior of the simulated airflows inside PG for each scenario presents behaviors with speeds less than 0.5 m/s (Figure 7). For the simulated scenario case S2—Hour 8, on the facades and the leeward and windward sides is where the airflow is most intense as well as velocity. The air that enters the greenhouse through these regions moves towards the central zone of PG where it rises towards the roofed zone looking to leave the interior of the structure through the ventilation areas located in this region. For this scenario, it is observed that the airflow patterns move in a series of converging cells between the perimeter areas of the greenhouse and the center of the greenhouse and between the soil zone and the PG roof zone (Figure 7).

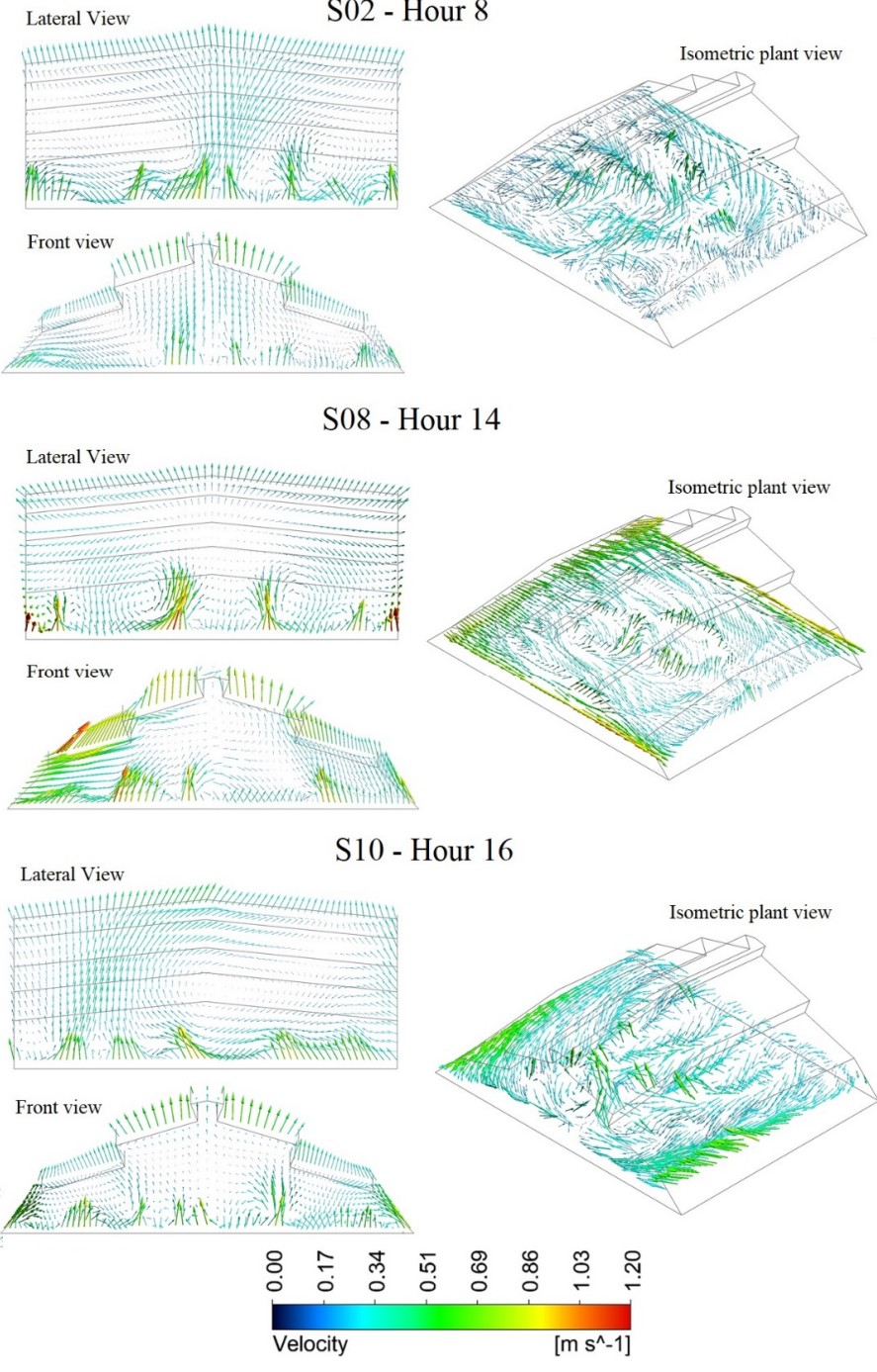

**Figure 7.** Simulated airflow patterns (m/s) inside the PG greenhouse.

For scenario S08—Hour 14, the characteristics of the airflow pattern described above are still observed, although for this case the velocity of the airflow vectors present a greater magnitude; this increase in velocity is caused by a higher velocity of the external airflow affecting the structure and a higher temperature inside PG due to higher radiation, which generates a greater acceleration of the airflow via free convection through the buoyancy phenomenon [39]. Likewise, it is observed that, for this condition part of the airflow that moves upwards inside the greenhouse returns and falls towards the ground zone generating a mixture with the air that enters PG through the sides, this effect is caused by the changes presented in the density of the air as a function of the variation of the temperature [66].

In the case of scenario S10—Hour 14, where the dominant wind direction is parallel to the ventilation areas of the roof region, it is possible to observe an airflow pattern that infiltrates the interior of the greenhouse through the leeward façade and is directed towards the central zone in movements with small variations in the upward and downward direction. It is also possible to observe that, near the average length of PG, the lateral areas are zones that allow the entrance of airflows towards the interior of PG. For this simulation in the middle region between the central zone and the leeward façade, we can observe the interaction of airflows that form an upward flow current directed towards the region of roof ventilation zones, this airflow is generated due to the restriction of airflow movement and loss of impulse caused by the presence of the anti-insect screen on the façade of the PG [39,67,68].

To determine the quantitative airflow behavior, simulated velocity data were extracted on the middle section of the structure along the X, Z axes at a height of 1.5 m above ground level and on the middle section of PG along the Y axis. For each set of data obtained in each axis, parameters such as the average speed ($V_{in}$) and the normalized speed ($V_{nor}$ = Velocity inside/Velocity outside) were calculated for each parameter with its respective standard deviation (Table 6). The $V_{in}$ values ranged from $0.21 \pm 0.09$ m/s to $0.36 \pm 0.13$ m/s for the X axis, from $0.15 \pm 0.08$ m/s to $0.37 \pm 0.12$ m/s for the Y axis, and from $0.26 \pm 0.08$ m/s to $0.46 \pm 0.18$ m/s.

**Table 6.** Mean velocity ($V_{in}$) and normalized velocity ($V_{nor}$) values obtained for each simulated scenario.

| X-Axis | | | | | |
|---|---|---|---|---|---|
| Scenario | $V_{in}$ (m/s) | $V_{nor}$ (%) | Scenario | $V_{in}$ (m/s) | $V_{nor}$ (%) |
| S01—Hour 7 | $0.26 \pm 0.14$ | $132 \pm 43.1$ | S07—Hour 13 | $0.33 \pm 0.15$ | $27.1 \pm 12.9$ |
| S02—Hour 8 | $0.21 \pm 0.09$ | $101 \pm 13.2$ | S08—Hour 14 | $0.32 \pm 0.13$ | $24.2 \pm 11.2$ |
| S03—Hour 9 | $0.24 \pm 0.07$ | $122 \pm 33.2$ | S09—Hour 15 | $0.35 \pm 0.13$ | $23.9 \pm 4.2$ |
| S04—Hour 10 | $0.32 \pm 0.11$ | $81.3 \pm 27.2$ | S10—Hour 16 | $0.31 \pm 0.11$ | $44.1 \pm 13.8$ |
| S05—Hour 11 | $0.36 \pm 0.13$ | $51.4 \pm 19.3$ | S11—Hour 17 | $0.32 \pm 0.12$ | $61.2 \pm 21.2$ |
| S06—Hour 12 | $0.35 \pm 0.17$ | $35.2 \pm 16.5$ | | | |
| Y-Axis | | | | | |
| Scenario | $V_{in}$ (m/s) | $V_{nor}$ (%) | Scenario | $V_{in}$ (m/s) | $V_{nor}$ (%) |
| S01—Hour 7 | $0.23 \pm 0.16$ | $119 \pm 67.1$ | S07—Hour 13 | $0.23 \pm 0.06$ | $19.5 \pm 5.6$ |
| S02—Hour 8 | $0.24 \pm 0.07$ | $121 \pm 34.2$ | S08—Hour 14 | $0.15 \pm 0.08$ | $12.3 \pm 6.5$ |
| S03—Hour 9 | $0.35 \pm 0.08$ | $161 \pm 35.1$ | S09—Hour 15 | $0.22 \pm 0.09$ | $19.1 \pm 8.3$ |
| S04—Hour 10 | $0.37 \pm 0.12$ | $44.3 \pm 28.2$ | S10—Hour 16 | $0.24 \pm 0.14$ | $33.5 \pm 21.8$ |
| S05—Hour 11 | $0.35 \pm 0.15$ | $51.1 \pm 21.3$ | S11—Hour 17 | $0.20 \pm 0.07$ | $61.2 \pm 13.2$ |
| S06—Hour 12 | $0.31 \pm 0.16$ | $32.1 \pm 18.2$ | | | |
| Z-Axis | | | | | |
| Scenario | $V_{in}$ (m/s) | $V_{nor}$ (%) | Scenario | $V_{in}$ (m/s) | $V_{nor}$ (%) |
| S01—Hour 7 | $0.29 \pm 0.10$ | $141 \pm 47.1$ | S07—Hour 13 | $0.40 \pm 0.18$ | $33.4 \pm 16.1$ |
| S02—Hour 8 | $0.26 \pm 0.08$ | $132 \pm 34.2$ | S08—Hour 14 | $0.46 \pm 0.18$ | $35.6 \pm 13.5$ |
| S03—Hour 9 | $0.32 \pm 0.18$ | $157 \pm 45.6$ | S09—Hour 15 | $0.29 \pm 0.11$ | $26.9 \pm 9.1$ |
| S04—Hour 10 | $0.33 \pm 0.14$ | $82.5 \pm 29.2$ | S10—Hour 16 | $0.30 \pm 0.09$ | $42.5 \pm 11.2$ |
| S05—Hour 11 | $0.43 \pm 0.15$ | $60.1 \pm 14.3$ | S11—Hour 17 | $0.27 \pm 0.08$ | $51.5 \pm 18.2$ |
| S06—Hour 12 | $0.44 \pm 0.17$ | $42.4 \pm 15.7$ | | | |

In the case of the normalized velocity ($V_{nor}$), it is observed that the scenarios S01, S02, and S03 that correspond to the hours 7, 8, and 9 respectively, hours of low wind speed outside (0.2 m/s) present values of velocity in the interior greater than those presented in the exterior; this can be an effect of the strong influence of the thermal effect on the natural ventilation for these scenarios. On the other hand, for the remaining scenarios, it was observed that the normalized velocity presented reductions with respect to the velocity of the exterior airflow of 19.7% and 79.6% for the X axis, 38.8% and 87.7% for the Y axis, and 17.5% and 73.1% for the Z axis.

Finally, the behavior of the normalized velocity was graphed through the length of each of the axes evaluated, where the variations in velocity already discussed can be seen that are presented in the scenarios S01, S02, and S03, and for the remaining scenarios it can be seen how the velocity of the airflows present more stable behaviors and with velocities lower than the speed of the external wind (Figure 8).

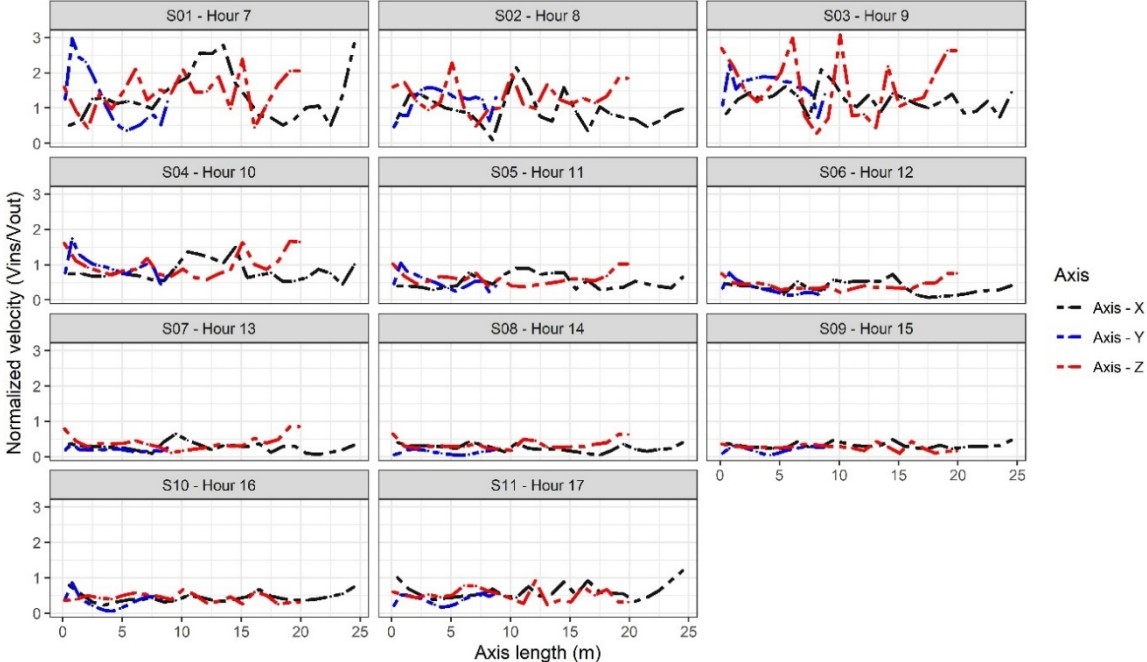

**Figure 8.** Normalized velocity (Vnor) distribution inside PG greenhouse.

*3.3. Spatial Temperature Distribution*

The spatial distribution of the temperature for scenario S02—Hour 8 (Figure 9) allows us to observe qualitatively how, inside the greenhouse, an increase in temperature begins to be generated in relation to the temperature of the outside environment. This is an effect generated by the passage of solar radiation through the translucent PG cover that allows energy to be stored in the form of heat in the inside environment [29,69]. For this scenario, it can be seen that the lower temperature regions are located just above the areas near the façades and the windward side walls, where there is a greater airflow; the warmer regions are located in the central region of the greenhouse with a trend towards the leeward side region (Figure 9).

In the case of the S08—Hour 14 scenario, a spatial distribution can be observed where the thermal gradients in the cross, longitudinal, and vertical axes are more qualitatively identifiable, with higher temperature values in the area of the PG cover and floor (Figure 9). Finally, for the scenario S10—Hour 16, a change in the spatial distribution is observed, caused by the direction of the external wind incident on the structure. For this scenario, it is observed how the region of lower temperature is located on the windward side, while the warm region is located just above the zone of interaction of the two

convective cells that are formed between the central region and the leeward side, which is a region of low air velocity.

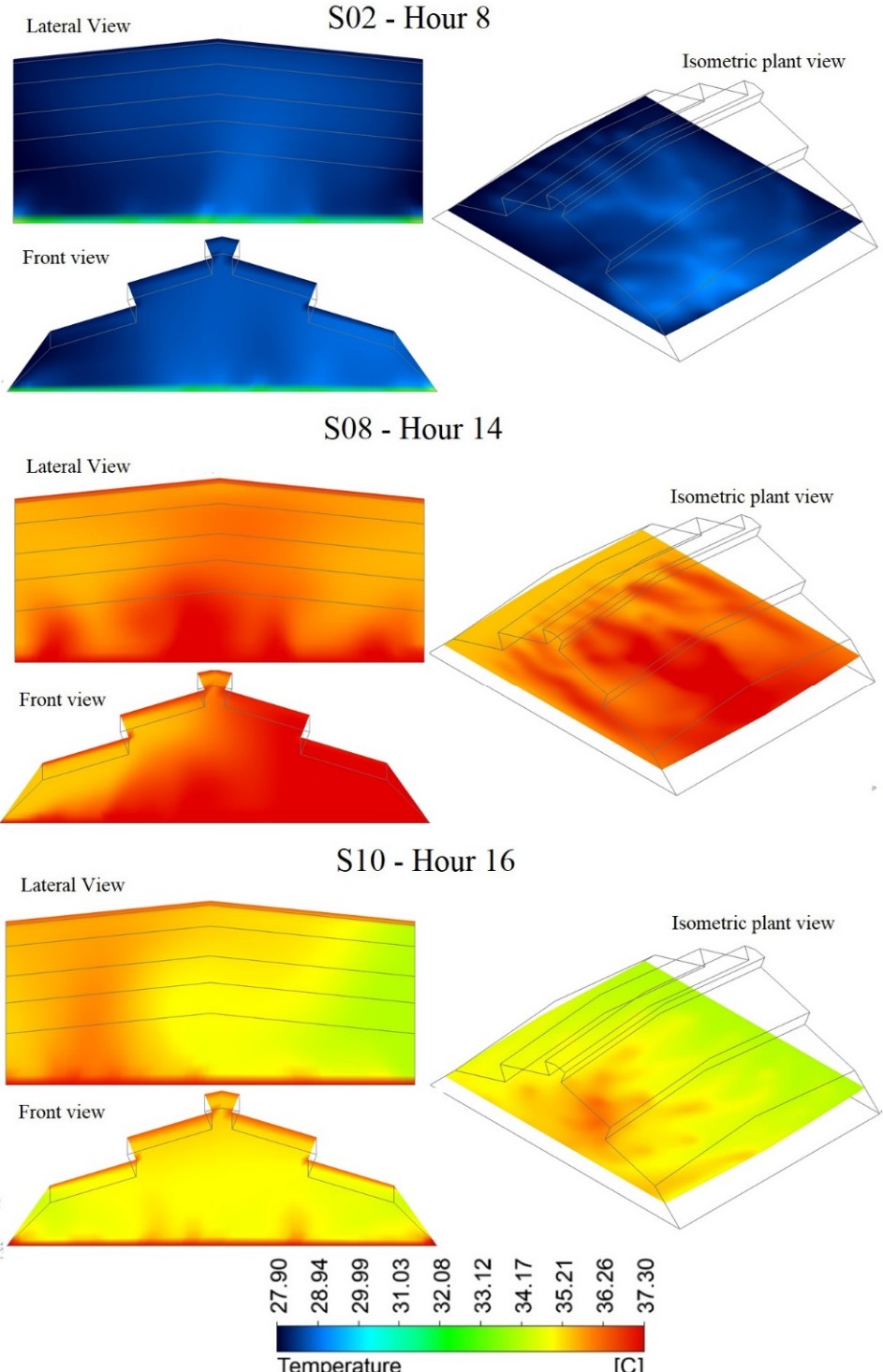

**Figure 9.** Simulated airflow patterns (m/s) inside the PG greenhouse.

For the quantitative analysis of this variable, the average values of the temperature inside the structure ($T_{in}$) and the average thermal differential ($\Delta T_m$ = Tin − Tout) were determined for each of the axes analyzed (Table 7). These results show that the value of the temperature inside the PG is a function of the temperature of the external environment and the level of solar radiation for each

scenario evaluated. Therefore, an increase in temperature between 7 and 14 h and a subsequent decrease until 17 h is observed, which is a characteristic phenomenon of passive greenhouses located in intertropical regions [70]. Tin values ranged from 25.4 ± 0.19 °C to 36.9 ± 0.13 °C for the X axis, from 25.6 ± 0.43 °C to 36.9 ± 1.69 °C for the Y axis, and from 25.5 ± 0.17 °C to 36.5 ± 0.51 °C for the Z axis.

**Table 7.** Average temperature ($T_{in}$) and average thermal gradient values ($\Delta T_m$) obtained for each simulated scenario.

| X-Axis | | | | | |
|---|---|---|---|---|---|
| Scenario | $T_{in}$ (°C) | $\Delta T_m$ (°C) | Scenario | $T_{in}$ (°C) | $\Delta T_m$ (°C) |
| S01—Hour 7 | 25.4 ± 0.19 | 0.44 ± 0.19 | S07—Hour 13 | 35.6 ± 0.24 | 0.42 ± 0.24 |
| S02—Hour 8 | 28.5 ± 0.26 | 0.70 ± 0.26 | S08—Hour 14 | 36.9 ± 0.13 | 1.56 ± 11.2 |
| S03—Hour 9 | 31.0 ± 0.39 | 0.90 ± 0.34 | S09—Hour 15 | 35.5 ± 0.21 | 0.65 ± 0.21 |
| S04—Hour 10 | 33.3 ± 0.77 | 1.40 ± 0.77 | S10—Hour 16 | 34.8 ± 0.20 | 0.73 ± 0.20 |
| S05—Hour 11 | 35.1 ± 0.97 | 1.62 ± 0.97 | S11—Hour 17 | 33.7 ± 0.15 | 0.67 ± 0.15 |
| S06—Hour 12 | 36.3 ± 0.82 | 1.48 ± 0.82 | | | |
| Eje Y | | | | | |
| Scenario | $T_{in}$ (°C) | $\Delta T_m$ (°C) | Scenario | $T_{in}$ (°C) | $\Delta T_m$ (°C) |
| S01—Hour 7 | 25.6 ± 0.43 | 0.70 ± 0.43 | S07—Hour 13 | 36.7 ± 1.61 | 1.50 ± 1.61 |
| S02—Hour 8 | 28.7 ± 0.55 | 0.91 ± 0.55 | S08—Hour 14 | 36.9 ± 1.69 | 1.52 ± 1.69 |
| S03—Hour 9 | 31.7 ± 0.86 | 1.62 ± 0.86 | S09—Hour 15 | 36.0 ± 1.06 | 1.12 ± 1.06 |
| S04—Hour 10 | 33.6 ± 1.42 | 1.71 ± 1.42 | S10—Hour 16 | 35.4 ± 1.15 | 1.28 ± 1.15 |
| S05—Hour 11 | 35.4 ± 1.69 | 1.91 ± 1.69 | S11—Hour 17 | 34.3 ± 0.90 | 1.18 ± 0.90 |
| S06—Hour 12 | 36.5 ± 1.66 | 1.67 ± 1.66 | | | |
| Eje Z | | | | | |
| Scenario | $T_{in}$ (°C) | $\Delta T_m$ (°C) | Scenario | $T_{in}$ (°C) | $\Delta T_m$ (°C) |
| S01—Hour 7 | 25.5 ± 0.17 | 0.58 ± 0.17 | S07—Hour 13 | 36.1 ± 0.37 | 0.93 ± 0.37 |
| S02—Hour 8 | 28.3 ± 0.26 | 0.49 ± 0.26 | S08—Hour 14 | 36.5 ± 0.51 | 1.06 ± 0.51 |
| S03—Hour 9 | 31.0 ± 0.47 | 0.89 ± 0.47 | S09—Hour 15 | 35.8 ± 0.69 | 0.86 ± 0.69 |
| S04—Hour 10 | 32.7 ± 0.37 | 0.82 ± 0.37 | S10—Hour 16 | 35.1 ± 0.60 | 0.95 ± 0.60 |
| S05—Hour 11 | 34.5 ± 0.42 | 1.02 ± 0.42 | S11—Hour 17 | 34.0 ± 0.65 | 0.90 ± 0.65 |
| S06—Hour 12 | 35.7 ± 0.36 | 0.86 ± 0.36 | | | |

Another relevant aspect that has aroused the interest of researchers in recent years is the study of spatial behavior in terms of homogeneity of variables of microclimate interest such as temperature and relative humidity inside greenhouses. It has been shown that there is a high heterogeneity in the value of these variables, which affects the growth and development of crops by spatially modifying physiological phenomena such as transpiration and absorption of nutrients by plants [71,72]. Therefore, to analyze the homogeneity of the temperature distribution, the value of ΔT was plotted for each hour through the length of the X, Y, and Z axes (Figure 10).

In general terms, it can be seen that, for the X and Z axes, the differences in ΔT in most scenarios remain below 2.0 °C, which can be considered as a homogeneous behavior since this is the maximum limit in terms of microclimatic uniformity that should exist between the highest and lowest temperature point within a protected agricultural structure [73].

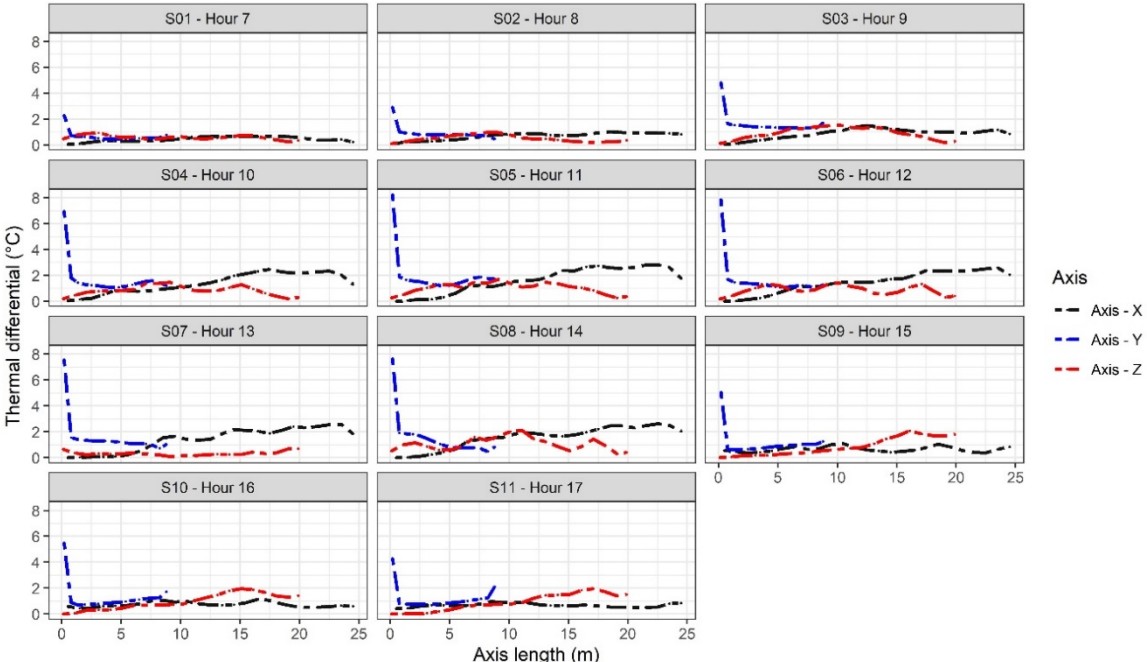

**Figure 10.** Thermal gradient distribution profiles (ΔTm) inside PG greenhouse.

### 3.4. Spatial Distribution of Relative Humidity

Relative humidity is another variable that generates interest in the management practices of the interior microclimate of the greenhouses. This variable under extreme levels of behavior promotes the appearance of diseases, affecting the growth and development of the crops, the health of the plants, and the final quality of the harvested products [55,74]. Extreme humidity conditions differentially affect crops under high temperature conditions, and humidity levels below 40% increase the plant's stomatic resistance and transpiration rate. Under this situation, if there is no water resource and an adequate irrigation system to meet the water needs demanded by the plant, the productivity of the crop will be limited by water stress conditions. On the other hand, humidity conditions with values higher than 85% can produce limited transpiration rates that generate dehydration, wilt, and necrosis in the leaves of the plants [75].

The spatial distribution patterns for relative humidity qualitatively allow us to observe that there are humidity gradients in each of the axes evaluated (Figure 11). In general terms, for the simulated scenarios, it can be observed that the distribution of relative humidity without the presence of crops is restricted to the psychrometric relationship of the humid air with respect to temperature; therefore, the patterns of spatial distribution are inverse to those of temperature, so that the drier regions coincide with the warmer regions and vice versa, and the areas with greater airflow present higher humidity values. Likewise, it was observed that the highest humidity is present in the areas with the greatest airflow, which coincides with what was reported by Teitel et al. [76].

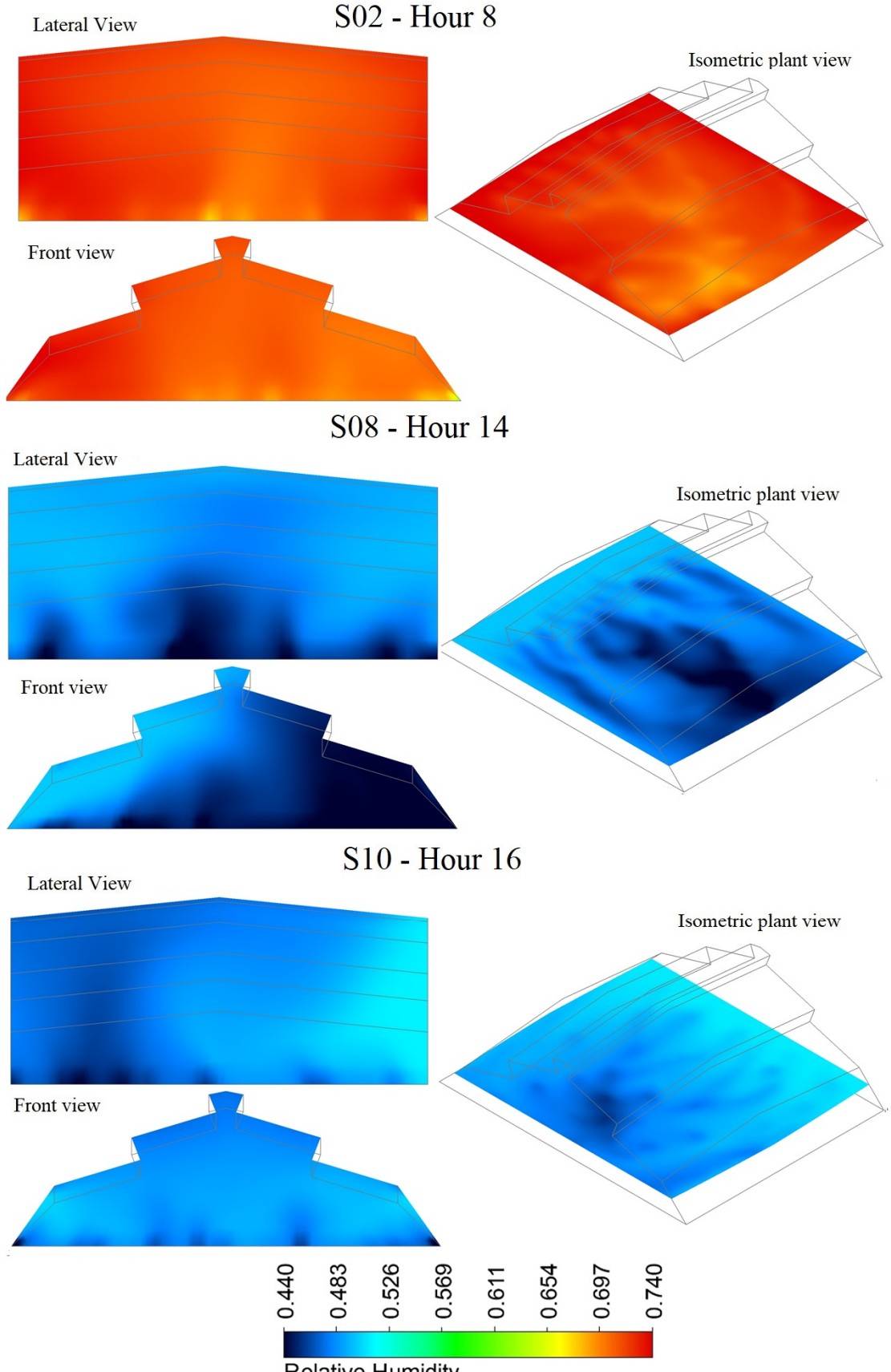

**Figure 11.** Patterns of spatial distribution of relative humidity (%) inside the PG greenhouse.

On the other hand, the average values of relative humidity inside the structure ($RH^{in}$) and the average relative humidity differential ($\Delta RH_m = RHin - RHout$) were calculated for each of the axes analyzed (Table 8). These values presented a temporal variation along the hours of the day as a function of the external relative humidity; therefore, it is observed that the highest humidity values were obtained for the first hours of the daytime period (7 and 8 h), then a general decrease of the humidity levels begins until reaching the minimum values above 14 h and from the following hour begins again an increase of the relative humidity value. $RH_{in}$ values ranged from 45.8 ± 2.14% to 77.8 ± 0.90% for the X axis, from 46.8 ± 1.40% to 77.1 ± 0.40% for the Y axis, and from 47.0 ± 1.31 to 77.1 ± 0.81% for the Z axis. This behavior generated negative values from $\Delta RH_m$ for all scenarios with magnitudes between −2.54% and −6.26%. These values can be considered low bought with other works where the humidity presents different values up to −13.1 °C [70]. The negative value indicates that the humidity level inside the PG is lower than that of the outside environment.

**Table 8.** Mean temperature values ($RH_{in}$) and mean thermal gradient ($\Delta RH_m$) obtained for each simulated scenario.

| | X-Axis | | | | |
|---|---|---|---|---|---|
| **Scenario** | **$RH_{in}$ (%)** | **$\Delta RH_m$ (%)** | **Scenario** | **$RH_{in}$ (%)** | **$\Delta RH_m$ (%)** |
| S01—Hour 7 | 77.8 ± 0.90 | −3.63 ± 0.90 | S07—Hour 13 | 47.4 ± 2.39 | −4.64 ± 2.39 |
| S02—Hour 8 | 71.3 ± 1.11 | −4.26 ± 1.11 | S08—Hour 14 | 45.8 ± 2.14 | −4.95 ± 2.14 |
| S03—Hour 9 | 64.2 ± 1.44 | −4.35 ± 1.44 | S09—Hour 15 | 47.6 ± 0.55 | −2.54 ± 0.55 |
| S04—Hour 10 | 56.7 ± 2.46 | −5.71 ± 2.46 | S10—Hour 16 | 48.9 ± 0.53 | −2.83 ± 0.53 |
| S05—Hour 11 | 50.9 ± 2.78 | −5.61 ± 2.78 | S11—Hour 17 | 51.6 ± 0.42 | −2.94 ± 0.42 |
| S06—Hour 12 | 48.1 ± 2.19 | −5.08 ± 2.19 | | | |
| | **Y-Axis** | | | | |
| **Scenario** | **$RH_{in}$ (%)** | **$\Delta RH_m$ (%)** | **Scenario** | **$RH_{in}$ (%)** | **$\Delta RH_m$ (%)** |
| S01—Hour 7 | 77.1 ± 0.40 | −4.29 ± 0.40 | S07—Hour 13 | 48.0 ± 0.75 | −4.09 ± 0.75 |
| S02—Hour 8 | 70.8 ± 0.49 | −4.79 ± 0.49 | S08—Hour 14 | 46.8 ± 1.40 | −3.94 ± 1.40 |
| S03—Hour 9 | 62.3 ± 0.48 | −6.26 ± 0.48 | S09—Hour 15 | 47.1 ± 0.42 | −3.09 ± 0.42 |
| S04—Hour 10 | 56.7 ± 0.96 | −5.69 ± 0.96 | S10—Hour 16 | 48.3 ± 0.56 | −3.51 ± 0.56 |
| S05—Hour 11 | 51.0 ± 0.76 | −4.60 ± 0.76 | S11—Hour 17 | 50.9 ± 0.57 | −3.68 ± 0.57 |
| S06—Hour 12 | 48.6 ± 0.77 | −4.09 ± 0.77 | | | |
| | **Z-Axis** | | | | |
| **Scenario** | **$RH_{in}$ (%)** | **$\Delta RH_m$ (%)** | **Scenario** | **$RH_{in}$ (%)** | **$\Delta RH_m$ (%)** |
| S01—Hour 7 | 77.1 ± 0.81 | −4.28 ± 0.81 | S07—Hour 13 | 48.6 ± 1.03 | −3.40 ± 1.03 |
| S02—Hour 8 | 72.2 ± 1.12 | −3.39 ± 1.12 | S08—Hour 14 | 47.0 ± 1.31 | −3.72 ± 1.31 |
| S03—Hour 9 | 64.2 ± 1.74 | −4.37 ± 1.74 | S09—Hour 15 | 47.1 ± 1.78 | −3.10 ± 1.78 |
| S04—Hour 10 | 58.5 ± 1.27 | −3.86 ± 1.27 | S10—Hour 16 | 48.4 ± 1.61 | −3.39 ± 1.61 |
| S05—Hour 11 | 52.6 ± 1.26 | −3.94 ± 1.26 | S11—Hour 17 | 51.0 ± 1.89 | −3.57 ± 1.89 |
| S06—Hour 12 | 49.7 ± 1.01 | −3.48 ± 1.01 | | | |

The spatial behavior of the value of ΔHR was plotted for each of the lengths of the axes evaluated (Figure 12). In general terms, the differences between the point of highest humidity and the point of lowest humidity inside the PG ranges from −2.5 to −7.5%. These ranges of variation can be considered acceptable since it is close to the recommended value of ±3% difference between the points of extreme relative humidity value inside a greenhouse [73].

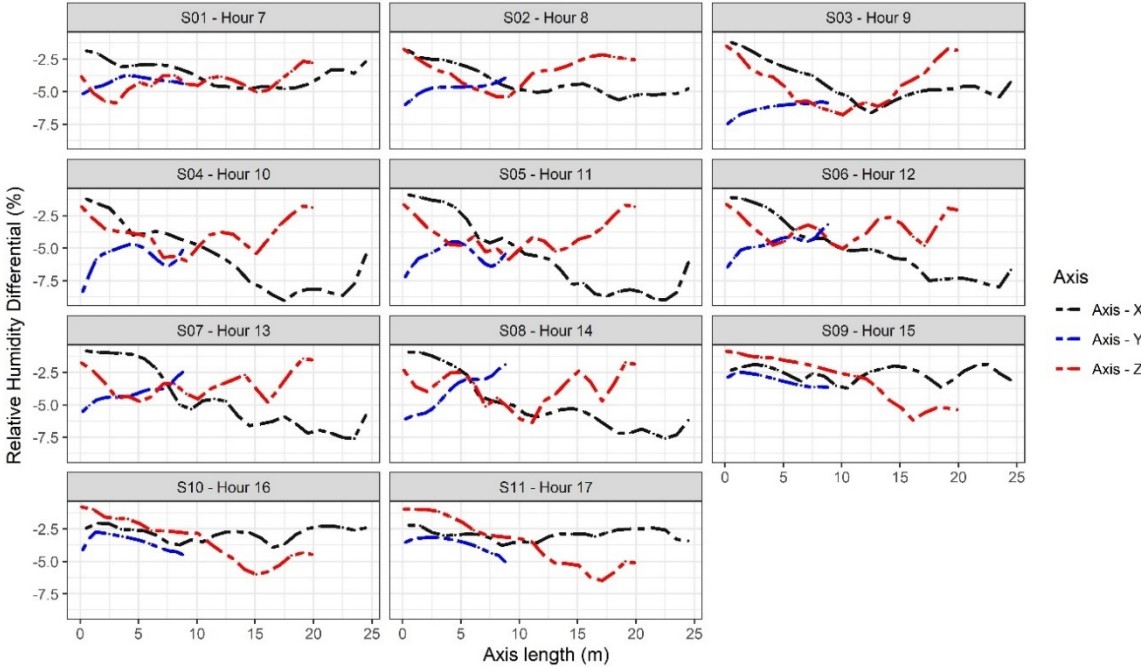

**Figure 12.** Relative humidity gradient distribution profiles (ΔRH) inside PG greenhouse.

## 4. Discussion

In this research, a 3D CFD numerical simulation model was implemented to study the natural ventilation of a passive greenhouse and its effect on the spatial distribution of temperature and relative humidity. One of the most relevant phases of numerical simulation is the experimental validation of the simulated process. The recording of climate variables inside the greenhouse on a real scale is one of the most widely used methodologies and can also be one of the most expensive depending on the size of the greenhouse [22]. In this case, it should be mentioned that the numerical model was partially validated for temperature and relative humidity, which does not detract from the validity of the results of this research, since there are a significant number of works in greenhouses with the same approach [77–80]. However, future studies should consider the validation of airflow patterns through sonic anemometry or wind tunnel studies.

The first stage of the validation of the numerical model corresponded to a statistical analysis aimed at validating or rejecting the use of the model as the main tool for the development of the objective of this research. The results obtained allowed establishing that the measured and simulated data sets did not present significant differences; therefore, there was no numerical evidence to establish that the CFD model was not able to predict the airflows and the microclimate inside the PG greenhouse. Subsequently, each of the data sets measured and simulated for each scenario studied was compared using goodness-of-fit parameters. Whereas, for the variable temperature, the values of MAE and RMSE presented similar values to those reported in other investigations of the study cap and that used the same methodological approach as those developed by Mesmoudi et al. [46], Saberian et al. [77], and Roman-Roldan et al. [63].

In the case of relative humidity, MAE and RMSE presented values slightly lower than those reported in the numerical study of natural ventilation for a passive greenhouse developed by Villagran and Bojacá. [55]. These goodness-of-fit values for humidity are below the recommended 10% error studies involving the use of simulation models in a greenhouse microclimate [81]. Finally, the existing correlation between measured and simulated data was established, finding that the data present $R^2$ values higher than 0.8, values that indicate a high correlation between the data sets. All these results allowed establishing that, in quantitative terms, the numerical model has a great capacity to predict the thermal and hygrometric behavior of the PG greenhouse under the specific conditions of

evaluation. Therefore, it was also possible to conclude that the CFD is a robust and versatile tool for the development of these types of studies, as has been recently mentioned by Li et al. [41].

Once the model was validated, the qualitative and quantitative results of the airflows were analyzed. It was possible to determine one of the hypotheses of the work and it is the one referring to that the qualitative behavior of the airflows inside PG is strongly influenced by factors such as the speed and direction of the external airflow in the greenhouse environment, as it had already been determined for other types of passive structures [38,82–84]. It was found that the presence of the porous insect screen in the ventilation areas of PG generates strong air interactions between the environment and outside near the perimeter of the greenhouse, which is consistent with that reported by Molina-Aiz et al. [85].

The airflows inside the greenhouse also showed a strong relationship with the thermal effect of the natural ventilation; most of these flows were directed vertically to the ventilation areas arranged in the region of the PG. This type of behavior is characteristic inside greenhouses when the thermal effect via buoyancy dominates the wind effect in the natural ventilation process, which generally occurs under conditions of low external wind speed (<1.5 m/s), such as those presented in the cases simulated in this research [86,87]. The airflow velocity values, obtained in each simulated scenario, present values lower than 0.5 m/s, which are velocity values very similar to those reported by Lopez et al. [88] in an experimental study of natural ventilation in a greenhouse equipped with anti-insect screens in the side and roof vents, which are characteristic of naturally ventilated greenhouses [89].

It is important to mention that these analyzed results are for low outside wind speeds (<1.5 m/s); therefore, future work can focus on the study and validation of the greenhouse model for outside wind conditions with higher speeds, evaluate other types of porous screens and their arrangement and installation over the ventilation areas since these parameters will directly influence the ventilation phenomenon and airflow patterns.

On the other hand, the spatial distribution of the temperature is directly related to the airflow patterns that occur inside PG; therefore, it is characteristic that, in naturally ventilated greenhouses, the regions with higher temperatures inside the greenhouse coincide with the regions with poor airflow and low velocity and vice versa [55,58]. Also, it was found that, for some simulated scenarios in the canopy and soil regions, there is a higher temperature value. This is due to the effect of the convective flow that occurs between these regions and the indoor and outdoor environments of the greenhouse and the fact that they are key factors determining the microclimate inside the greenhouse PG [58,90].

Likewise, in the regions where two or more convective movement cells interact over the central area of the greenhouse, a short circuit or zone of low air movement is produced, which translates into a region of higher temperature, which coincides with what was reported by Sase [66] and by Molina-Aiz et al. [26]. In quantitative terms, it could be observed that Tin values between 11 and 14 h exceed the 35 °C recommended value to ensure plant growth and development in warm regions [77,91], although it should be noted that the temperature behavior in PG does not exceed 40 °C, as reported in another series of studies for warm regions [77].

With regard to the fact that values of $\Delta T_m$ generated were below 2.0 °C in all scenarios, which is very relevant for this type of climate conditions, this value is much lower than values reported in other studies where values of $\Delta T_m$ are reached between 5.0 °C and 10.2 °C, for greenhouses equipped with insect-proof screens [58,92]. Finally, it is observed that the greatest thermal differentials occur in the axis Y. Where the highest temperature values occur near the floor, values that decrease as they approach the ventilation areas located on the roof of PG, this is consistent with the results obtained by Majdoubi et al. [93].

Finally, for the relative humidity it was identified that its spatial distribution was inverse to the spatial distribution of the temperature; therefore, regions of high temperature in turn are regions of low relative humidity and vice versa. This is in accordance with what was reported by Villagran and Bojaca [70]. The average relative humidity values inside the greenhouse were between 47% and 78%. Values that do not exceed the 80% value set as a high humidity limit and also are not below the 40%

limit are considered a low humidity limit [75]. This can be considered as acceptable since, in principle, the crops would not be subjected to any kind of stress due to inadequate humidity conditions, which can affect the growth and development of the plants [74].

## 5. Conclusions

The CFD model implemented in this research proved to be an agile, robust, and reliable tool that allowed to satisfactorily predict the thermal and hygrometric behavior of the evaluated greenhouse. The validation of the model was satisfactory through statistical analysis and by comparing goodness-of-fit parameters between measured and simulated data.

It was demonstrated that the spatial distribution of temperature and relative humidity in this type of greenhouse is a function of the flow patterns generated inside the structures, patterns that, in turn, depend on the conditions of speed and direction of outside wind, the geometry of the greenhouse and the presence of anti-insect screens in the ventilation areas.

In this study, it was found that the gradients between indoor and outdoor environment did not present values higher than 2.0 °C for temperature and 6.3% for relative humidity, values that are relatively lower than those reported in other researches worldwide where temperature gradients higher than 10.2 °C and for humidity with values higher than 13.1% have been found.

This research work, despite its limitations in the validation process and its modeling and simulation simplifications by not considering any type of crop inside the greenhouse, provides results and an approach that can be the reference for future work aimed at microclimatic optimization of such structures that certainly contribute to improving the sustainability of food production systems and improve levels of food security globally.

Finally, it is important to highlight the relevance of this work for the area of knowledge of protected agriculture, mainly in aspects such as the use of a new prototype of greenhouse design for a warm region and the study of the spatial distribution of relative humidity through numerical simulation, a microclimate parameter that has been little studied under this methodological approach.

**Author Contributions:** Conceptualization, E.V.; methodology, E.V., R.L., A.R., and J.J.; software, E.V.; validation, E.V., R.L., A.R.; investigation, E.V., R.L., and J.J.; writing—original draft preparation, E.V., R.L., and J.J.; writing—review and editing, E.V., R.L., A.R., and J.J. All authors have read and agreed to the published version of the manuscript.

**Funding:** The research was funded by The Regional Fund of Agricultural Research and Technological Development (FONTAGRO) as part of the project "Innovations for horticulture in protected environments in tropical zones: an option for sustainable intensification of family farming in the context of climate change in LAC". The opinions expressed in this publication are solely those of the authors and do not necessarily reflect the views of FONTAGRO, its Board of Directors, the Bank, its sponsoring institutions, or the countries it represents

**Acknowledgments:** The authors wish to thank the Corporación Colombiana de Investigación Agropecuaria (AGROSAVIA) for their technical and administrative support in this study.

**Conflicts of Interest:** The authors declare no conflict of interest. The funders had no role in the design of the study; in the collection, analyses, or interpretation of data; in the writing of the manuscript, or in the decision to publish the results.

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
