# Peer review of "3D Numerical Analysis of the Natural Ventilation Behavior in a Colombian Greenhouse Established in Warm Climate Conditions"

_sustainability, doi:10.3390/su12198101_

Round 1

Reviewer 1 Report

More detail about the simulation methodology, and choice of simulation parameters will be good. A discussion of the sensitivity of the validation to the various input model parameters will be helpful as well, i.e. would different choices of parameters have resulted in equally good match to the experimental results?

Author Response

Appreciated.

Reviewer.

Thank you for your comments and suggestions. You will find the answers in the archive.

Greetings

Reviewer 2 Report

The article „3D Numerical Analysis of the Natural Ventilation Behavior in a Colombian Greenhouse Established in Warm Climate Conditions“ it has a certain scientific value. However, some improvement is needed. In the following paragraphs I present a few observations that should be taken into account in the revised version of manuscript.

Section “Abstract”:

- The abstract needs to be shortened and sharpened. Remove the ballast.
- at the end of the abstract there is no clearer assessment of the global impact conclusion

Section “Introduction”:

- research hypotheses are missing at the end of the introduction. It is necessary to set hypotheses as it is standard in scientific work. Hypotheses need to be verified or refuted in conclusions.

- at the end of the introduction, it is necessary to define more clearly the objectives of this research and its need in an international context. Specifically, who will benefit from it.

Section “Methods”:

- recommend complementing the statistical analysis of research data in the research. The statistical analysis will further enhance the research results. Statistical analysis should be part of every scientific work.

- it is necessary to add what has been compared in more detail so that the methodology can be applied purely in the future
- to characterize what hypothesis was verified by what statistical method

Section “conclusion”:

- separate the conclusion section from the discussion

- the established hypotheses need to be upheld.
-the conclusion is to be conceived both locally and globally

- the conclusion should be sharpened and the ballast removed as it is too lengthy

General comments:
- The article lacks a substantial chapter which should be called "discussion". It needs to be supplemented and discussed with authors and work in this area. The discussion should be part of every scientific work.
- the discussion needs to be completed to have a global reach

- In the discussion I recommend to discuss with already published articles in "this journal" and etc. journals, especially with those dealing with similar issues. For example, these and more:

https://doi.org/10.1590/2447-536X.v26i2.2149; DOI: https://doi.org/10.26444/aaem/81314 ; https://doi.org/10.1016/j.scienta.2016.01.030; DOI: 10.1016/S2095-3119(19)62598-0; DOI: https://doi.org/10.26444/aaem/80908 and etc.

-the end must be sharpened and shortened. Remove the ballast.

I suggest major revision. After removing the shortcomings, I would like to re-examine the manuscript and reconsider my position.

Author Response

Appreciated.

Reviewer.

Thank you for your comments and suggestions. You will find the answers in the archive.
In general all were added to the new version of the manuscript and you will be able to identify them because they are highlighted in yellow.

Greetings

the authors

Round 2

Reviewer 2 Report

I satisfied.